# Innovative Surface-Enhanced Raman Spectroscopy Method as a Fast Tool to Assess the Oxidation of Lipids in Ground Pork

Magdalena Wrona [1], Juliette Lours [2], Jesús Salafranca [1,*], Catherine Joly [2] and Cristina Nerín [1]

1   Instituto de Investigación en Ingeniería de Aragón (I3A), Departamento de Química Analítica, Escuela de Ingeniería y Arquitectura (EINA), Universidad de Zaragoza, María de Luna 3 (Edificio Torres Quevedo), 50018 Zaragoza, Spain

2   Laboratoire de Bioingenierie et Dynamique Microbienne aux Interfaces Alimentaires (BioDyMIA, EA n∘3733), ISARA Lyon, Université de Lyon, IUT Lyon 1, Technopole Alimentec, Rue Henri de Boissieu, F–01000 Bourg en Bresse, France

*   Correspondence: fjsl@unizar.es; Tel.: +34-976-762-628

**Abstract:** A novel method for the determination of lipid oxidation using Raman microscopy was developed. A home-made surface-enhanced Raman spectroscopy (SERS) substrate based on silver nanoparticles deposited on a glass Petri dish was used. The degradation of ground pork stored at 5 °C was monitored for 16 days. Two packages were considered: an active packaging containing an oregano extract with antioxidant properties and a conventional one consisting of a low-density polyethylene (LDPE) film. The lipid fraction of the ground pork was extracted with a mixture of diethyl ether/n-hexane (1:1). A remarkable maximum signal enhancement factor of $1.64 \times 10^7$ at 1439 cm$^{-1}$ shift (and up to $8.58 \times 10^6$ at 1655 cm$^{-1}$, chosen for fat oxidation assessment) was obtained with SERS compared to conventional Raman. In addition, SERS provided better discrimination among samples than the results obtained by the thiobarbituric acid reactive substances (TBARS) method. The experimental conditions for SERS were optimized and discussed.

**Keywords:** lipid oxidation; surface-enhanced Raman spectroscopy (SERS); silver nanoparticles; Raman microscopy





## 1. Introduction

Over the past few decades, consumer concern about the freshness and quality of meat increased significantly [1]. Meat and meat products are an example of food products that are susceptible to deterioration due to their composition [2]. In addition to the bacterial spoilage, the main degradation process of ground pork is caused by protein oxidation, which is characterized by loss of red color and lipid oxidation, associated with off-flavors and off-odors [3,4].

Lipid peroxidation occurs via free radical chain reactions, with reactive oxygen species (ROS) such as hydroxyl (OH·) and hydroperoxyl (OOH·) radicals being the major initiators [5]. Several factors such as fat content, storage temperature, kind of meat, oxygen availability, and type of packaging can greatly influence the extent of lipid oxidation [6]. Since the half-life of ROS is extremely short, their direct measurement is not an easy task. Several methods based on the detection of different fat oxidation products are used. One of the most common is the thiobarbituric acid reactive substances (TBARS) assay. In this case, malondialdehyde (MDA), considered to be one of the main end products of lipid peroxidation, already present in the meat is measured, as well as that generated by the hydrolysis reaction. However, several considerations must be taken into account. First, not all fats produce MDA. Second, other substances may be generated by fat peroxidation. In addition, some food additives may react with MDA, thus underestimating the final result. Finally, thiobarbituric acid can react with other food components [7,8]. For these

reasons, and despite its popularity, TBARS results must be considered and interpreted with caution [9].

Recent advances in active food packaging allow for the extension of the shelf life of fresh foods [10–14]. Most researchers focused their work on delaying the lipid oxidation of fatty foods and meats by adding active agents such as antioxidants to the packaging materials [15–18]. Therefore, a new simple method for the determination of lipid oxidation would be highly appreciated, not only because fatty acids are important for the nutritional value of meat (especially the polyunsaturated ones, such as linoleic acid, around 11.8% and linolenic acid, 1.0% in pork), but also to evaluate the efficiency of the new active packaging in the preserved food.

Raman spectroscopy can be considered as a useful tool for such purposes. Its main advantages are: little or no sample preparation, very weak scattering of water and carbon dioxide, rapid analysis, selectivity, versatility and the possibility of qualitative and quantitative analysis of both organic and inorganic compounds. On the other hand, limited sensitivity and high detection limits are relatively common. To overcome these disadvantages, surface-enhanced Raman spectroscopy (SERS) is a technique in which a special substrate, such as surfaces covered with roughened metals, or colloidal particles greatly enhances the Raman signal. The most common SERS substrates are made of silver, gold, or copper [19] and allow enhancement factors (EF) up to $10^{11}$ [20,21]. SERS allows both quantitative and qualitative analysis of chemical species. Today, there are two reliable theories that explain the SERS phenomena, by considering either electromagnetic or chemical mechanisms [21,22].

Conventional Raman spectroscopy was already applied to evaluate lipid oxidation [23], to discriminate and characterize oils [24–27] and to control fat content or changes in meat [28,29], with mixed results. Although some SERS studies focused on lipids [30–32], to the best of our knowledge, SERS was never used as a method for the determination of lipid oxidation in meat or meat products.

In this work, a different approach to evaluate fat degradation is presented. For the first time, a home-made SERS substrate, much cheaper than commercial ones, was developed and used for the analysis of lipids in ground meat. Silver nanoparticles (AgNPs) were deposited in situ on glass Petri dishes by means of a "silver mirror" reaction. The degradation of ground meat stored at 5 °C was monitored for 16 days. Two packaging systems were used to evaluate differences in lipid oxidation: an active packaging containing an oregano extract with antioxidant properties and a conventional packaging consisting of a low-density polyethylene (LDPE) film used as a reference. The results were also compared with the TBARS method. The experimental conditions were optimized and discussed.

## 2. Materials and Methods

### 2.1. Reagents

Trichloroacetic acid (≥99%, CAS 76-03-9), thiobarbituric acid (98%, CAS 504-17-6), 1,1,3,3-tetraethoxypropane (≥96.0%, CAS 122-31-6), silver nitrate (≥99.0%, CAS 7761-88-8), and nitric acid (70%, CAS 7697-37-2) were purchased from Sigma-Aldrich Química S.A. (Madrid, Spain). Hydrochloric acid (37%, CAS 7647-01-0), ammonium hydroxide solution (32%, CAS 1336-21-6), ethanol (≥99.9%, 64-17-5), n-hexane (analytical grade, CAS 110-54-3), and diethyl ether (≥99.7%, CAS 60-29-7) were from Sharlau (Barcelona, Spain). Potassium sodium tartrate tetrahydrate (≥99%, CAS 7697-37-2) was from Merck Millipore (Madrid, Spain). Ultrapure water was obtained from a Wasserlab Ultramatic GR system (Barbatáin, Spain).

### 2.2. Samples

Samples of fresh ground pork were purchased in bulk from a local supermarket. For this study, the use of packaged ground pork was not recommended because the ingredient lists reported the addition of vegetable extracts including carrots and spices. These extracts

contained lycopene, a red pigment, which can be easily extracted along with the lipids, causing serious interferences in the Raman spectra.

### 2.3. Films

Polymeric films were prepared and supplied by Artibal S.A. (Sabiñánigo, Spain). The active films consisted of low-density polyethylene (LDPE) coated with an active varnish containing oregano extract as antioxidant, produced under patent EP1477519 A1. Blank reference films of conventional LDPE, without active coating but with the same structure, were also supplied.

### 2.4. Sample Preparation

For each sample, 22 g of meat was placed in 5 cm diameter polystyrene Petri dishes. Then, the meat was covered with active film ($6 \times 6$ cm) and packed in a LDPE bag ($7 \times 9$ cm) under normal atmosphere using an impulse sealer PFS-200 Zhejiang Dongfeng Packing Machine Co (Wenzhou, Zhejiang, China). Reference samples were also prepared using conventional LDPE film. After being packaged, samples were refrigerated at 5 °C, and they were analyzed after 0, 7, 9, 11, 14, and 16 days.

### 2.5. Extraction of Lipids

The lipid extraction method developed in our previous research was applied [33]. Briefly, 12 g of meat was accurately weighed and extracted three times with a mixture of n-hexane/diethyl ether (1:1, *v/v*). After combining the three fractions together, the extract was evaporated to dryness in an R-124 rotary evaporator with a B-480 water bath from Büchi (Flawil, Switzerland), redissolved in 5 mL of n-hexane, and carefully evaporated under nitrogen stream until 1 mL. After deposition in the silvered glass Petri dish and evaporation of the solvent, SERS measurement of the fatty extract was carried out. All the experiments were performed in triplicate and all the measurements were performed three times.

### 2.6. SERS Substrate

Based on a well-known procedure for silvering glass [34], 15 mL of aqueous solution of silver nitrate at concentration 6.2% (*w/v*) was prepared and a small part, about 1 mL, was reserved. Then, 32% ammonium hydroxide solution was added drop by drop to the $AgNO_3$ solution until the chocolate-colored precipitate was just dissolved. Then, the remaining mL of $AgNO_3$ was added, showing an incipient turbidity. The resulting solution was diluted to 100 mL with ultrapure water and stored in a brown glass bottle. Second, the reducing solution was prepared as follows: 0.19 g of potassium sodium tartrate was added to 100 mL of boiling water. Then, 20 mL of an aqueous solution of $AgNO_3$ at a concentration of 1.1% (*w/v*) was slowly added while stirring vigorously. The solution was boiled for 10 min and then cooled to room temperature. The resulting solution was filtered through a cellulose acetate syringe filter (25 mm diameter, 0.50 μm pore size) and stored in a brown glass bottle.

To prepare the SERS substrate, 2 mL of silver nitrate solution and 2 mL of the reducing solution were added to a soda lime glass Petri dish (40 mm diameter, 12 mm height). The solutions were mixed and the Petri dish was left for the deposition of silver nanoparticles (AgNPs) which should start after approximately 20 s. The initial reaction time was used as an indicator of the freshness of the solutions. In our experience, they should be renewed every month to ensure optimal performance. After 10 min of reaction, the Petri dish was washed with ultrapure water and rinsed with ethanol. Finally, a gentle stream of nitrogen was applied to completely dry the surface. Figure 1a shows a finished silvered Petri dish, and the structure of the AgNPs was verified by scanning electron microscopy (see Figure 1b).

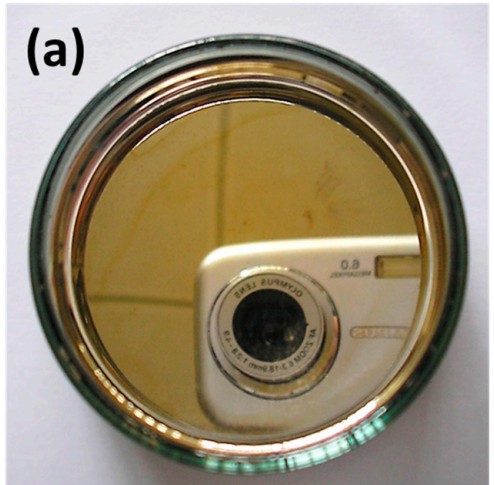
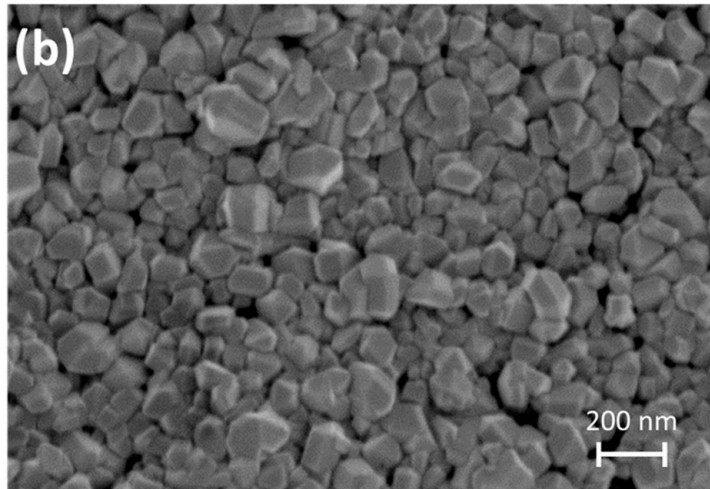

**Figure 1.** (**a**) Petri dish with SERS substrate, (**b**) SEM micrograph of SERS substrate.

### 2.7. Equipment

The Raman spectra were obtained using a DXR™ Raman spectrometer equipped with a microscope from a Thermo Scientific (Waltham, MA, USA). In addition, a motorized microscope stage sample holder and an excitation laser source at 532 nm were used, while the detector consisted of a charge-coupled device. The Raman microscope was first automatically aligned and calibrated to ensure and maintain optimal results. Both neon and polystyrene standards were used in the calibration process with the tool provided with the system. The following conditions were used for SERS measurements: $10\times$ objective (0.4 NA), aperture was 25 μm pinhole, the grating was 900 lines·mm$^{-1}$, and the spectral range was 3500–50 cm$^{-1}$. The optimal laser power was set to 5 mW, and 20 exposures of 30 s each were selected to acquire all spectra. The following options were applied: white light correction, fluorescence correction, cosmic ray rejection, and a smart background option. The camera temperature was −51 °C. Omnic v. 9.2 software from Thermo Fisher Scientific was used for data collection and analysis. Spectra were compared using the common scale mode.

Scanning electron microscopy images of the silver substrate were obtained with a Carl Zeiss (Jena, Germany) MERLIN™ field emission scanning electron microscope (FE-SEM) with an accelerating voltage of 20.0 kV.

Absorbance of TBARS samples was measured using a Shimadzu UV1700 Pharmaspec spectrophotometer (Kyoto, Japan).

### 2.8. Cooling System for Improved Raman Measurement

In our previous study, on the application of SERS for the determination of butylated hydroxyanisole in edible and essential oils [35], a special cooling system was developed to perform analyses at low temperature. Significant differences between room and low temperature were obtained due to the presence of very intense Raman shifts belonging to the oil, a complex matrix. In short, a Thermo-Haake K20 refrigerated bath with a C10 immersion circulator was used. A stainless steel block with two 8 mm holes was connected to the circulator by Tygon® tubing covered by air-conditioning pipe insulation. A Petri dish containing the oil samples was placed on the block, which was placed on the microscope stage. Raman spectra were recorded at −2 °C. Figure 2 shows a diagram of the system.

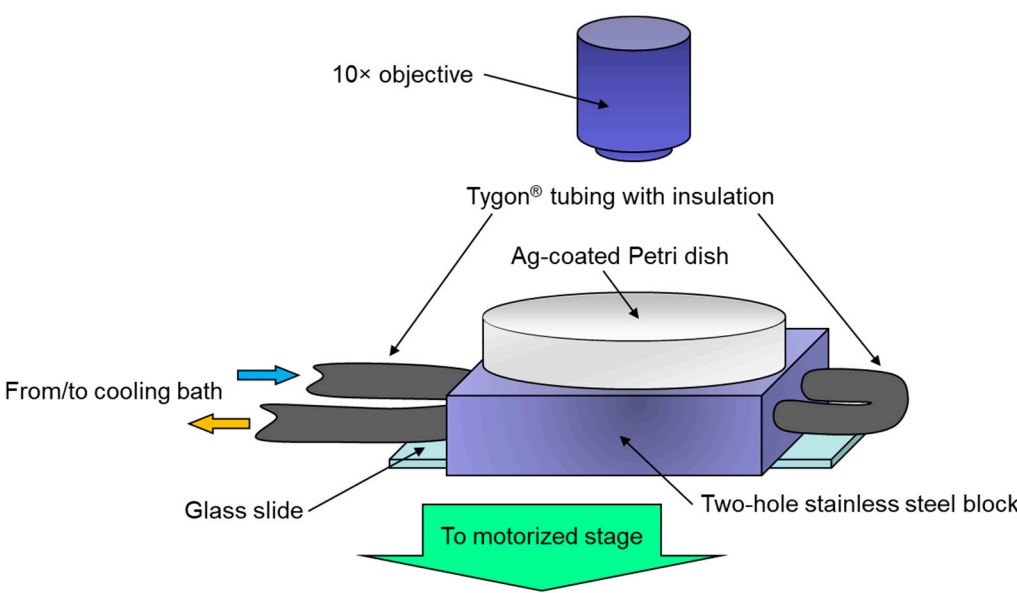

**Figure 2.** Diagram of the home-made cooling system.

### 2.9. TBARS

The lipid oxidation results obtained by SERS were compared with those obtained by the TBARS method [36]. Briefly, 20 mL of an aqueous solution of trichloroacetic acid at a concentration of 10 $\mu g \cdot g^{-1}$ was added to 10 g of ground pork. An IKA (Staufen, Germany) Ultra-Turrax at 18,000 rpm was used to grind the mixture until a uniform slurry was obtained. The supernatant was then filtered through 90 mm diameter Whatman grade 5 qualitative filter paper from Sigma Aldrich (Madrid, Spain). 2 mL of a 20 mM aqueous of thiobarbituric acid solution was added to 2 mL of the filtered aliquot and mixed. The mixture was then kept in a silicone oil bath at 97 °C for 20 min. Absorbance was measured at 532 nm against a blank. The results were expressed in equivalent concentration of MDA (mg of malondialdehyde·$kg^{-1}$ of meat). The MDA solution was prepared from 1,1,3,3-tetraethoxypropane dissolved in aqueous HCl 1 M. Concentrations from 0.1 to 0.8 $\mu g \cdot g^{-1}$ of the MDA solution were used for the calibration curve.

### 2.10. Optical Images of Samples

A Nikon Coolpix 4300 (Tokyo, Japan) digital camera was used to photograph meat samples at a height of 15 cm and at an angle of 45°. The samples were illuminated from above. Manual mode with flash cancel was used. White balance was adjusted manually using a white chroma meter standard. The following settings were used: macro close-up as focus mode, fine image quality, image size of 1600, sensitivity in ISO auto, and metering in matrix mode.

### 2.11. Statistical Analysis

All results presented are expressed as mean ± standard deviation. Student's *t*-test was used to evaluate the significant differences between the TBARS and SERS results. The null hypothesis was applied based on the similarity of the samples. The difference between the samples was significant (significance level $\alpha = 0.05$) when the experimental value of t was greater than the tabulated value. In this case, the null hypothesis was rejected.

In addition, Grubb's test was used to test for outliers. When the calculated value of G ($\alpha = 0.05$) exceeded the critical value, the suspect value was rejected.

## 3. Results and Discussion

It is worth noting that the SERS substrate obtained by AgNPs deposition resulted in a smooth surface without defects. Moreover, the scanning electron microscope micrograph (Figure 1b) showed a uniform surface of small nanoparticles (~150 nm) with homogeneous

size distribution as result of the "silver mirror" reaction. The precision of the mean particle size, expressed as %RSD, for three independent substrates was 14.2%.

First, the weight of meat was optimized to obtain a significant fraction of the extracted lipids, since the amount of fat deposited on the SERS substrate was crucial. On the one hand, a drop of lipid extract was completely evaporated after laser heating. On the other hand, when a thick layer (8 mm) was applied, no SERS effect was observed at all. Finally, a layer of about 3 mm of fat was used, which corresponds to about 12 g of meat.

The next parameter to be optimized was the measurement temperature. No significant differences were found when comparing fresh and oxidized samples at room temperature. Taking into account the results obtained from our previous work [35], when using the cooling device at $-2\,^\circ$C, better definition of the relevant bands and notorious differences in intensity appeared in the oxidized sample, and so, further analyses were performed at low temperature. Although the effect of temperature on SERS was poorly explored, our findings were consistent with a work [37] suggesting that the dielectric constant of noble metal nanoparticles is influenced by electron–phonon scattering and electron–electron scattering, with EF being favored at low temperatures. In addition, laser power (ranging from 3.5 to 9.5 mW), exposure time, and number of exposures were adjusted by sequential optimization. The selected conditions were as follows: 5 mW of laser power, 30 s of exposure time, and 20 exposures. Finally, the results of area and height of the spectral bands were collected. Since significant differences were obtained only in the case of areas, this parameter was used in subsequent analyses.

An analysis of the photographs of meat samples showed that active packaging containing oregano extract prolonged the shelf life of meat. Oregano extract is a well-known antioxidant that was widely studied for active packaging purposes [10,38–43]. Therefore, it was chosen as antioxidant agent to inhibit the degradation of lipids in meat. Figure 3 shows the aspect of meat samples stored under refrigeration after 5 days. LDPE-packaged samples started to lose color due to myoglobin oxidation. This process was shown to be associated with lipid oxidation [4].

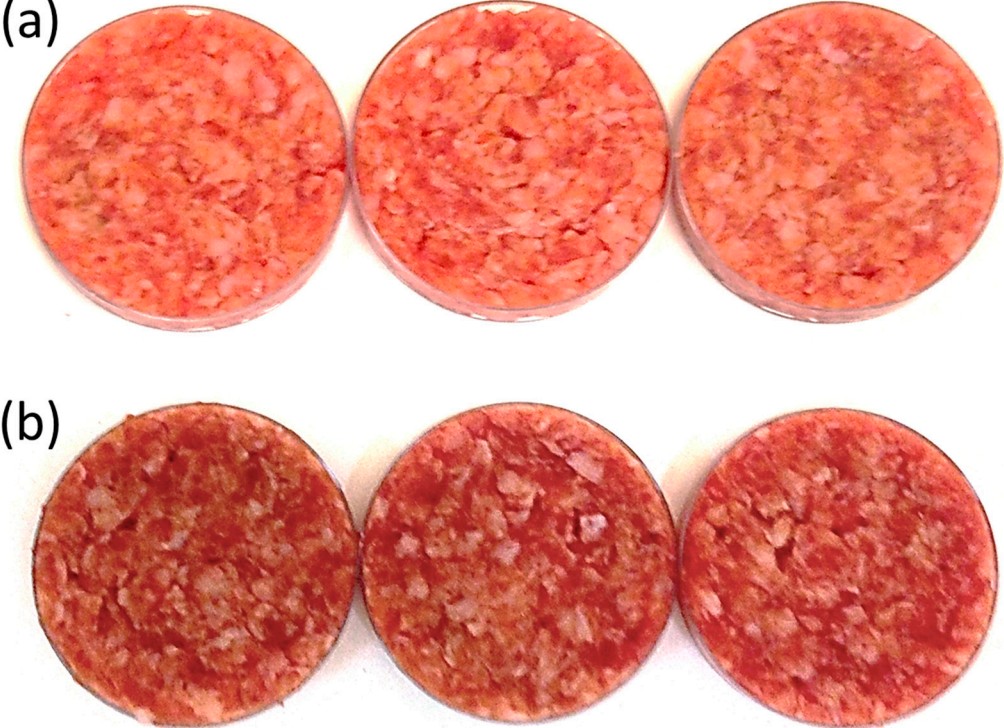

**Figure 3.** Aspect of meat samples stored at 5 $^\circ$C after 5 days, where: (**a**) samples from conventional LDPE packaging, (**b**) samples from active packaging.

In this work, the Raman spectrum of fat extracted from meat stored in active packaging was collected and compared with the spectrum of fat extracted from meat stored in conventional LDPE packaging. Figure 4 shows the spectrum of the extracted lipids at 0 days, highlighting the major Raman shifts associated with fat oxidation in the SERS spectrum. As can be seen, the SERS spectrum showed a much improved signal-to-noise ratio compared to conventional Raman.

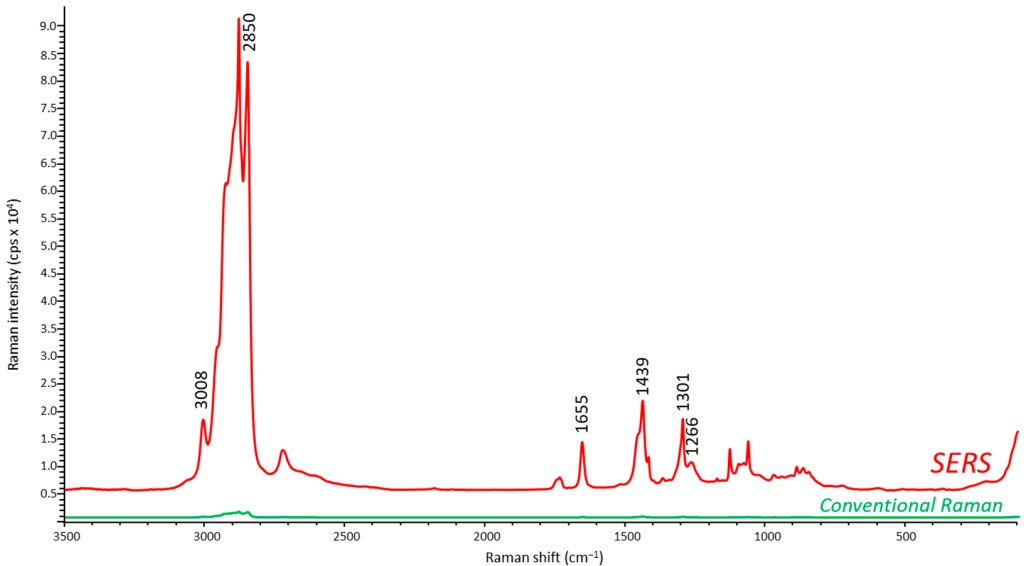

**Figure 4.** Spectrum of lipids extracted from ground pork after 0 days by SERS (red) and conventional Raman (green). Both spectra are shown on the same scale.

Raman shifts corresponding to the oxidation of lipids present in oils and fats were extensively described in the scientific literature [44–48]. The most relevant detected shifts related to lipid degradation are briefly described in Table 1.

**Table 1.** Raman shifts responsible for lipid oxidation obtained by SERS.

| Raman Shift [cm$^{-1}$] | Functional Group |
| --- | --- |
| 3008 | C–H stretching of C=C–H |
| 2850 | C–H stretching $\nu$(C–H) in –CH$_3$ and –CH$_2$– |
| 1655 | stretching of C=C bonds |
| 1439 | scissoring deformation $\delta$(CH$_2$) of –CH$_2$– |
| 1301 | in-plane C=C–H deformation |
| 1266 | in-phase –CH$_2$– twisting |

Three replicates of each sample were measured and three measurements of each replicate were performed in the case of SERS. The RSD% values for the measurements of lipids extracted from ground pork in LDPE and in active packaging, were less than 29% for LDPE and 15% for active packaging.

The SERS bands showed strong enhancement with respect to the ordinary Raman spectrum. In this work, the shifts at 1439 cm$^{-1}$ and 1655 cm$^{-1}$ were selected because they gave the highest values of the EF. According to calculations [49] (considering the refractive index of the fat as 1.46 [50], an average molecular weight of 272.27 Da, according to the fatty acid composition of pork meat [51], and a fat density of 0.91 g·cm$^{-3}$ [52]), the dimensions of the hourglass-shaped sample volume were ~2.6 μm diameter spot in the focal plane with a depth of ~30 μm. By subtracting the volume of silver nanoparticles, the effective Raman sampling of fat was about 82 μm$^3$. Considering the characteristics of the fat extract, the detection volume contained ~0.88 pmol.

The EFs were calculated from an analytical chemistry point of view [53,54] according to the formula $EF = (I_{SERS}/N_{SERS})/(I_{Raman}/N_{Raman})$, where $I_{SERS}$ and $I_{Raman}$ were the signal intensities with and without silver substrate, respectively, $N_{SERS}$ the number of molecules responsible for the enhancement (first layer in contact with silver, $\sim 8.4 \times 10^6$) and $N_{Raman}$ the number of fat molecules in the detection volume ($\sim 5.3 \times 10^{11}$). The maximum EF values obtained were $1.64 \times 10^7$ at 1439 cm$^{-1}$ and $8.58 \times 10^6$ at 1655 cm$^{-1}$. These Raman shifts were chosen as reference for normalization (1439 cm$^{-1}$, CH$_2$ scissoring, which is relatively unaffected by oxidation) and for assessing oxidative fat degradation, which was mainly due to unsaturation (1655 cm$^{-1}$, C=C stretching). Taking into account possible interferences, MDA, one of the main end products of lipid peroxidation, was considered as a reference. In a recent study [55], the MDA signal by SERS with mixed Au/Ag nanoparticles in the vicinity of 1655 cm$^{-1}$, more precisely at 1663 cm$^{-1}$ used by the authors for the effective measurement of a complex between MDA and 4-aminophenylthiophenol, was negligible. Since lipid peroxidation occurs mainly in unsaturated fatty acids, and double bonds between carbon atoms are lost in the process, the decrease in the peak at 1655 cm$^{-1}$ can be satisfactorily attributed to lipid peroxidation. Figure 5 shows the overlapped spectra on different days showing both bands, corresponding to samples stored with LDPE (left) and with active packaging (right). It can be clearly seen that the decrease in the 1655 cm$^{-1}$ band over time was much less pronounced in meat samples stored in active packaging. This demonstrates the protective effect of active agent by delaying fat oxidation.

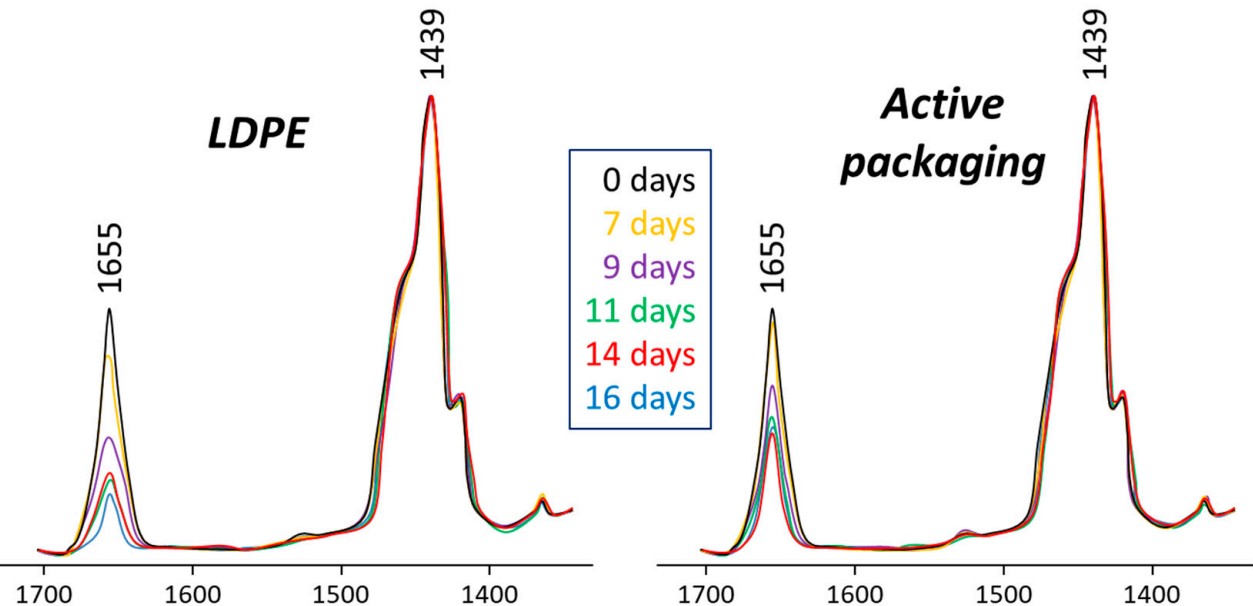

**Figure 5.** Superimposed Raman spectra on different days of meat samples stored in conventional LDPE (**left**) and active packaging (**right**) showing the decrease in the 1655 cm$^{-1}$ band (C=C stretching) compared to the normalized 1439 cm$^{-1}$ band (CH$_2$ scissoring) chosen as reference.

Figure 6 shows the graphical results of the SERS analysis, where a significant difference between conventional and active packaging was observed, as the error bars did not overlap. Therefore, the developed method can be successfully applied to monitor fat oxidation or to study the role of active packaging on fat and fatty products. Furthermore, a decreasing trend was observed from day 7 onwards. Again, since the active packaging always had a higher ratio area between 1655 and 1439 cm$^{-1}$ bands compared to LDPE-packaged samples, it can be confirmed that the active packaging provided an effective reduction in unsaturation of fats as a consequence of their oxidation.

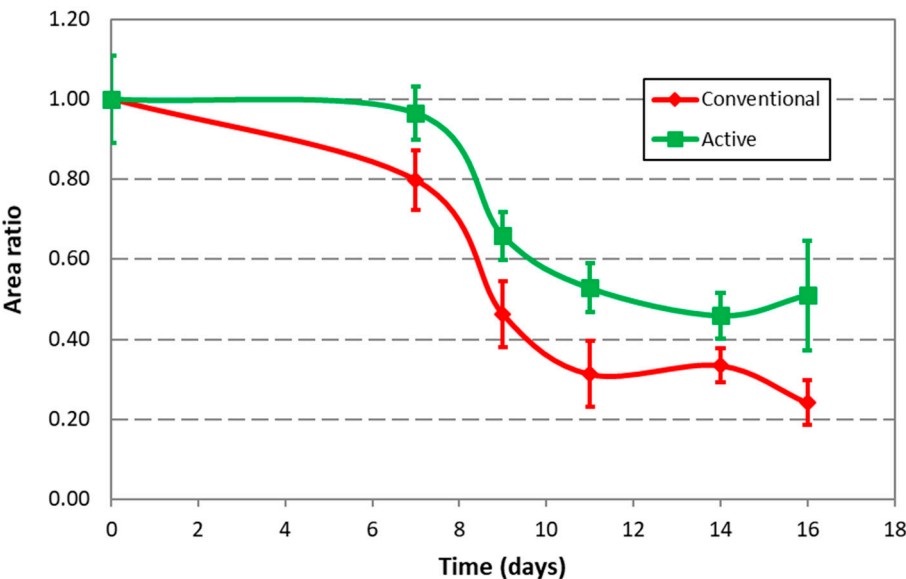

**Figure 6.** Results of SERS analysis of meat samples from LDPE and active packaging expressed as average area ratio ($1655/1439$ cm$^{-1}$ shifts) $\pm$ standard deviation, $n = 3$.

The quantitative analysis of unsaturation was performed as follows: first, the protective effect of the active packaging was evaluated by dividing the area ratio of bands at $1655/1439$ cm$^{-1}$ of conventional LDPE vs. active packaging samples. The lower the value, the better the performance of the active packaging in preventing lipid oxidation. In addition, the relative change of unsaturation (RCU%) over time was calculated for each series of samples according to the equation: RCU% = $((AR_0 - AR_n)/AR_0) \times 100$, where $AR_0$ and $AR_n$ are the area ratios between the 1655 and 1439 cm$^{-1}$ shifts corresponding to day 0 and after "$n$" days of storage, respectively. In this case, the higher the value, the more pronounced the oxidative degradation of the fat. All the results, expressed in percentages, are shown in Table 2. The results over time showed a significant efficacy of the active packaging. In addition, the RCU% values indicated an increase in saturation between samples of the same type with increasing storage time. The results always showed lower RCU% values for active packaging samples compared to LDPE-packaged meat.

**Table 2.** Results of unsaturation analysis of meat samples based on the SERS measurement of area ratio between 1655 and 1439 cm$^{-1}$ bands. All results are expressed as mean $\pm$ standard deviation (in percentage).

| Day | RCU% (LDPE) | RCU% (Active Packaging) |
| --- | --- | --- |
| 0 | $0 \pm 0$ | $0 \pm 0$ |
| 7 | $19 \pm 3$ | $1 \pm 0$ |
| 9 | $51 \pm 9$ | $30 \pm 3$ |
| 11 | $66 \pm 9$ | $44 \pm 5$ |
| 14 | $71 \pm 12$ | $51 \pm 10$ |
| 16 | $74 \pm 14$ | $49 \pm 10$ |

The quantification of MDA was performed by external calibration based on the measurement of absorbance. The linearity of the calibration curve was obtained in the range of 0.09–0.81 µg·g$^{-1}$ nominal concentrations with regression coefficient (r) 0.9990. Figure 7 shows the obtained values of TBARS for the meat samples stored in different packages during the 16 days of storage.

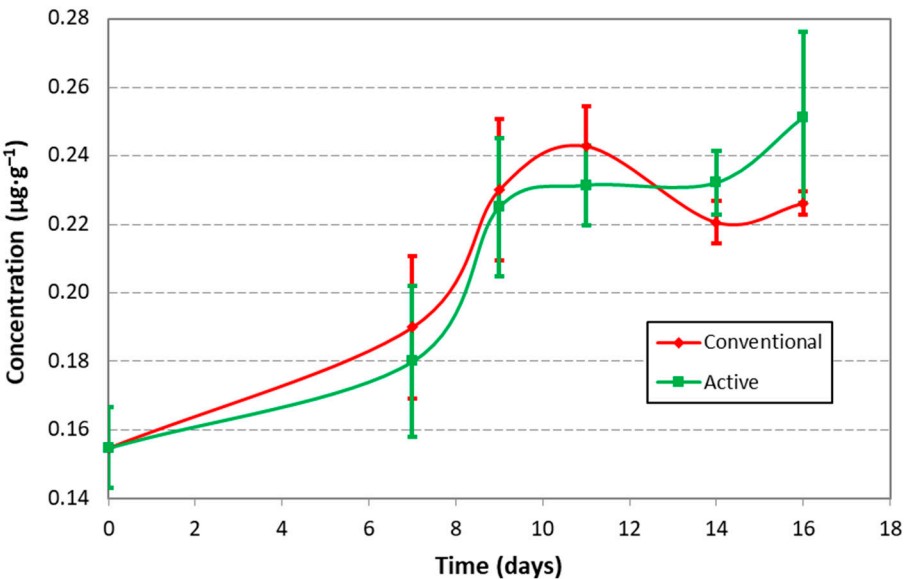

**Figure 7.** Results of TBARS of meat samples from LDPE and active packaging expressed as average concentration of MDA ± standard deviation, *n* = 3.

An increasing trend was observed compared to the day 0 samples. As expected, lipids underwent oxidation with the subsequent production of TBARS during the storage time, although the rate of TBARS formation varied among samples. The RSD% ranged from 11% (conventional LDPE) to 17% (active packaging samples). The *t*-test was used to compare the average values of the LDPE stored samples and those with active packaging. It was clear that the TBARS results were inconclusive and the inherent limitations of the method (some fats do not produce MDA, some food additives may react with MDA leading to an underestimation of results, and thiobarbituric acid may react with other food components) were noted.

### 4. Conclusions

A novel application of SERS using a home-made silver substrate and a cooler for improved measurement of samples was presented. The determination of lipid oxidation in packaged ground pork was performed. The proposed method was fast, with minimal handling of the fat sample. Better discrimination than the TBARS method to follow lipid degradation in short time was successfully achieved. The detection of differences between conventional and active packaging was confirmed. A cooling setup was shown to improve the sensitivity of SERS and allowed the study of a complex matrix. Furthermore, an outstanding SERS enhancement factor of more than $10^7$ was obtained compared to conventional Raman.

**Author Contributions:** Conceptualization, M.W., J.S. and C.J.; methodology, M.W., J.L. and J.S.; software, M.W., J.L. and J.S.; validation, M.W. and J.S.; formal analysis, M.W., J.L. and J.S.; investigation, M.W., J.L. and J.S.; resources, J.S. and C.N.; data curation, J.S. and C.N.; writing—original draft preparation, M.W., J.L., J.S., C.J. and C.N.; writing—review and editing, M.W. and J.S.; visualization, M.W. and J.S.; supervision, M.W. and J.S.; project administration, J.S. and C.N.; funding acquisition, J.S., C.J. and C.N. All authors have read and agreed to the published version of the manuscript.

**Funding:** This research received no external funding.

**Institutional Review Board Statement:** Not applicable.

**Informed Consent Statement:** Not applicable.

**Data Availability Statement:** The data presented in this study are available on request from the corresponding author.

**Acknowledgments:** Warm and sincere thanks are given to Thermo Fisher Scientific for the fruitful collaboration and effective exchange of experience and knowledge. M. Wrona acknowledges the FPU grant (reference number AP2012-2716) received from the MEC, Ministerio de Educación, Cultura y Deporte, Spain. J. Lours acknowledges the IUT Lyon 1 through the scholarship EXPLO'RA sup of the Region Rhône-Alpes. We also thank Gobierno de Aragón and Fondo Social Europeo for the financial help to GUIA T53_20R. The authors would like to acknowledge the use of Servicio General de Apoyo a la Investigación-SAI, Universidad de Zaragoza.

**Conflicts of Interest:** The authors declare no conflict of interest.

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
