# Peer review of "Innovative Surface-Enhanced Raman Spectroscopy Method as a Fast Tool to Assess the Oxidation of Lipids in Ground Pork"

_applsci, doi:10.3390/app13095533_

Round 1

Reviewer 1 Report

Comments to the authors

 The reviewer’s remarks are highlighted in yellow in the file

 The Introduction is well written. The characteristics and mechanism of lipid peroxidation are presented logically and coherently.

Row 51: The reviewer recommends that the authors either describe the meaning of the fatty acids (or specific fatty acids) mentioned in this case or delete/rewrite this part of the sentence.

 Material and methods: the applied methods are suitable for this study.

Row 102: The reviewer recommends on the authors to write the full name at least once outside the abstract (where this is not recommended) and then use the abbreviation in the text.

 Results and Discussion: Are relatively well written.

 Row 277: Fig.5 - there is a discrepancy between the text under the figure 5 and the labels on the graph. Clarify - does it refer to day 0 only or does it represent the changes observed over the 16 days?

Appropriate statistical methods are used to evaluate and compare the results obtained by different analysis.

 Conclusion:

The conclusions are well written.

 References

Over than 54% of the cited literature is relatively old - before 2015. Presumably there is more contemporary literature? If there is - let it be included.

 Minor editing of English language required

Author Response

RESPONSE TO REVIEWER #1

  • Following your indications highlighted in yellow in the file, “ground pork” and “thiobarbituric acid reactive substances (TBARS)” have been deleted from keywords. In addition, your comment “please describe in more details the mentioned (or observed) limitations of the method in this case” (lines 364-366 in the Word file with “all the revisions” selected), has been supplemented with a short explanation “(some fats do not produce MDA, some food additives may react with MDA leading to an underestimation of the results, and thiobarbituric acid may react with other food components)” similar to the introduction (lines 42-50).
  • Row 51: The reviewer recommends that the authors either describe the meaning of the fatty acids (or specific fatty acids) mentioned in this case or delete/rewrite this part of the sentence.

Response: The explanation “(especially the polyunsaturated ones, such as linoleic acid, around 11.8% and linolenic acid, 1.0% in pork)” has been inserted in lines 56-57.

  • Row 102: The reviewer recommends on the authors to write the full name at least once outside the abstract (where this is not recommended) and then use the abbreviation in the text.

Response: In section 2.3 “low density polyethylene (LDPE)” has been included.

  • Row 277: Fig.5 - there is a discrepancy between the text under the figure 5 and the labels on the Clarify - does it refer to day 0 only or does it represent the changes observed over the 16 days?

Response: Sorry for the mistake, the caption of Figure 5 was incorrect and now the correct text is displayed as “Superimposed Raman spectra on different days of meat samples stored in conventional LDPE (left) and active packaging (right) showing the decrease of the 1655 cm–1 band (C=C stretching) compared to the normalized 1439 cm–1 band (CH2 scissoring) chosen as reference.”

  • References: Over than 54% of the cited literature is relatively old - before 2015. Presumably there is more contemporary literature? If there is - let it be included.

Response: Where possible, references have been updated. In particular, the references 2, 3, 5, 6, 9, 28, 29 and 38 have been updated with more recent works. In addition, two new references (37 and 55) have been added.

We greatly appreciate the reviewer's comments and hope that our explanations and modifications to the manuscript will live up to his/her expectations.

Reviewer 2 Report

Please replace Fig. 1a with a clearer image.

Figure captions of Fig. 4 and Fig. 5 are the same, please revise.

Line 241-242, “The SERS spectrum showed a much improved signal-to-noise ratio”. Please provide the ordinary Raman spectrum, and compare it with SERS spectrum.

The homogeneity and reproducibility of SERS substrates need to be evaluated.

Line 93-94, “The use of packaged ground pork was highly inadvisable because of the addition of vegetable extracts including carrots and spices.” Does this mean that this method cannot be used for the testing of commercially packaged meat products? What is the scope of application of this method?

Line 219-222, “no significant differences were found when comparing fresh and oxidized samples at room temperature. However, when using the cooling device at –2 oC, additional Raman shifts appeared in the oxidized sample.” Please explain why.

Author Response

RESPONSE TO REVIEWER #2

  • Please replace Fig. 1a with a clearer image.

Response: A new image has been taken with more color and definition. We hope that this is now satisfactory.

  • Figure captions of Fig. 4 and Fig. 5 are the same, please revise.

Response: Thanks for pointing this out. Indeed, it was a mistake that has been corrected. The correct text of figure 5 is now displayed as “Superimposed Raman spectra on different days of meat samples stored in conventional LDPE (left) and active packaging (right) showing the decrease of the 1655 cm–1 band (C=C stretching) compared to the normalized 1439 cm–1 band (CH2 scissoring) chosen as reference.”

  • Line 241-242, “The SERS spectrum showed a much improved signal-to-noise ratio”. Please provide the ordinary Raman spectrum, and compare it with SERS spectrum.

Response: Figure 4 has been modified to include the conventional Raman spectrum in green below the SERS spectrum to show the huge signal differences between the two. The corresponding caption has been changed to “Figure 4. Spectrum of lipids extracted from ground pork after 0 days by SERS (red) and conventional Raman (green). Both spectra are shown on the same scale”. In addition, the preceding text (lines 257-260 in the Word file with “all the revisions” selected) has been slightly modified to read: “Figure 4 shows the spectrum of the extracted lipids at 0 days, highlighting the major Raman shifts associated with fat oxidation in the SERS spectrum. As can be seen, the SERS spectrum showed a much improved signal-to-noise ratio compared to conventional Raman”.

  • The homogeneity and reproducibility of SERS substrates need to be evaluated.

Response: Calculations have been made and the information is now included in the text (lines 220-221): “The precision of the mean particle size, expressed as %RSD, for 3 independent substrates was 14.2%.”

  • Line 93-94, “The use of packaged ground pork was highly inadvisable because of the addition of vegetable extracts including carrots and spices.” Does this mean that this method cannot be used for the testing of commercially packaged meat products? What is the scope of application of this method?

Response: When designing the study, we considered the possibility of interferences. After consulting the ingredient lists of various packaged ground pork products, we searched the spectra libraries provided with the Raman microscope and found similarities in the possible regions of analytical interest. Here you can see the conventional Raman overlap spectra of lard (red, possible interfering bands 1463 & 1742 cm-1) and lycopene (blue, 1458 & 1720 cm-1).

So we decide to avoid the unwanted contributions to the peroxidation process as much as possible for a first study. Does this mean that our method cannot be applied to packaged products? Not at all. In fact, we plan to carry out this precise study in the future to identify possible limitations, including the possibility of exploring other shifts if necessary, even at the cost of sacrificing sensitivity with lower EF.

In any case, the sentence has been qualified and the text (lines 98-100) now reads: “For this study, the use of packaged ground pork was not recommended because the ingredient lists reported the addition of vegetable extracts including carrots and spices.”

  • Line 219-222, “no significant differences were found when comparing fresh and oxidized samples at room temperature. However, when using the cooling device at ‑2 oC, additional Raman shifts appeared in the oxidized sample.” Please explain why.

Response: The original sentence has been qualified (lines 230-232), and the following text, including a new reference that we hope will be satisfactory to the reviewer, has been inserted at lines 233-236. “Although the effect of temperature on SERS is poorly explored, our findings are consistent with a work [37] suggesting that the dielectric constant of noble metal nanoparticles is influenced by electron-phonon scattering and electron-electron scattering, with EF being favored at low temperatures.”

We greatly appreciate the reviewer's comments and hope that our explanations and modifications to the manuscript will live up to his/her expectations.

Reviewer 3 Report

The Manuscript entitle ‘Innovative surface-enhanced Raman spectroscopy method as a fast tool to assess the oxidation of lipids in ground pork’. The MS provides some interesting insights into lipid peroxidation determination by SERS. Overall, the manuscript effectively summarizes the findings of the study and provides additional insights into the implications of your results. Just one comment, please explain the possible mechanism of the lipid change combined with the SERS analysis. This will help to understand the reasons for the change in peak better.

Author Response

RESPONSE TO REVIEWER #3

  • Just one comment, please explain the possible mechanism of the lipid change combined with the SERS analysis. This will help to understand the reasons for the change in peak better.

Response: Thank you very much for your kind comments. The following text, including a new reference from Wu et al. (2022) has been added to lines 296-303 in the Word file with “all the revisions” selected. “Taking into account possible interferences, MDA, one of the main end products of lipid peroxidation, has been considered as a reference. In a recent study [54], the MDA signal by SERS with mixed Au/Ag nanoparticles in the vicinity of 1655 cm–1, more precisely at 1663 cm–1 used by the authors for the effective measurement of a complex between MDA and 4-aminophenylthiophenol, was negligible. Since lipid peroxidation occurs mainly in un-saturated fatty acids, and double bonds between carbon atoms are lost in the process, the decrease of the peak at 1655 cm–1 can be satisfactorily attributed to lipid peroxidation”. We hope this is satisfactory for the reviewer and useful to readers.

We greatly appreciate the reviewer's comments and hope that our explanations and modifications to the manuscript will live up to his/her expectations.

Round 2

Reviewer 2 Report

The author replied to all questions, I have no additional questions.